# What Is Quality Food? The Opinion of Key of the Brazilian Food System

**DOI:** 10.3390/nu16070948

**Published:** 2024-03-26

**Authors:** Thaíse Gomes, Suellen Secchi Martinelli, Panmela Soares, Suzi Barletto Cavalli

**Affiliations:** 1Nutrition Postgraduate Programme, Federal University of Santa Catarina, Florianópolis 88040-900, Brazil; nutrithaisegomes@gmail.com (T.G.); suellen.smartinelli@gmail.com (S.S.M.); sbcavalli@gmail.com (S.B.C.); 2Nutrition Department, Federal University of Santa Catarina, Florianópolis 88040-900, Brazil; 3Department of Community Nursing, Preventive Medicine and Public Health and History of Science, University of Alicante, 03690 Alicante, Spain

**Keywords:** quality, food system, food and nutritional security

## Abstract

The aim of the study was to explore the concept of quality food in the opinion of key informants of the food system. This qualitative research included 208 key informants related to the food supply for Brazilian public food services. The participants were grouped into three groups according to their participation in the food system: 1. Food production; 2. Management and marketing; 3. Meal’s production process. Key informants answered the following question: “In your opinion, what is quality food?”. The answers were analysed through qualitative content analysis. The data analysis resulted in 52 codes grouped into eight categories, expressing the opinion of the study participants about what quality food is: nutritional, sustainable, sensory, hygienic–sanitary, care, regulatory, dependability and symbolic. Nutritional and sustainable dimensions were predominantly considered. The activities carried out in the food system seem to influence the opinion on food quality. The groups linked to food production put more emphasis on the sustainable dimension, followed by the nutritional dimension, while the groups linked to management and marketing and to the meals production process put more emphasis on the nutritional and sensory dimensions. These differences may indicate a difficulty in the transition towards a more sustainable and healthy food system.

## 1. Introduction

When it comes to food, the term “quality” is considered complex [1], relative and also multidimensional [2], encompassing several dimensions. These dimensions consist of a set of characteristics attributed to foods, such as nutritious, safe, tasty, good-looking and intact [2,3,4,5]. In general, these are dimensions that involve aspects of quality from the entire spectrum of food provision and food consumption, such as nutritional, symbolic or cultural, sanitary, regulatory, sensory [3,6,7], ease of use and eco-friendliness [6,7].

The several concepts of quality food can be informed by different influences related to culture [1] and individual food experiences during food acquisition, preparation and consumption [8]. Episodes resulting from the lack of food safety can also influence the formation of different concepts of food quality for consumers [9]. As a result, the safety requirements for foods imposed by consumers have influenced companies to focus on product quality [10]. Other issues also affect the formation of different points of view on aspects that may be related to food quality [10], such as the use of pesticides [11], food waste, and [12] public health problems such as obesity and malnutrition [13].

Individual experiences with food can influence expectations and opinions about its quality [1,2,5,14]. In this sense, the activities that take place in the food system, from cultivation to consumption [15], can influence opinions about food quality. In turn, how those involved in the system perceive quality can directly influence the quality of the food that is produced and offered [2]. Thus, the development of interventions aimed at building a more sustainable and healthier food system, capable of providing food that encompasses the different dimensions of quality, can be facilitated by recognising what these people consider to be quality food.

Quality food has been the focus of several studies. However, these studies focus on food consumers [16,17,18,19,20,21,22]. There are few studies aimed at identifying the concept of quality food in the opinion of those involved in the food system. When they exist, they seek to identify the opinion on the quality of a specific food [23], and not an overview. The aim of this study is to explore the concept of quality food in the opinion of key informants linked to different stages of the food system.

For this purpose, the study involved key informants involved in the school feeding services and in the Popular Restaurant Programme (from agricultural production to the provision of meals). The National School Feeding Program is a public policy that aims to provide healthy meals to all students enrolled in the public school network [24]. The program is present in all municipalities in the country and is managed by the education councils. The Popular Restaurant Program is part of the network of Public Food and Nutrition Facilities and aims to provide healthy meals to the population in situations of food insecurity, especially in large Brazilian cities.

## 2. Materials and Methods

### 2.1. Study Design

An exploratory and descriptive qualitative study was conducted by analyzing the content of open interviews with 208 key informants who played different roles in the meals production for Brazilian public food services (school meals and popular restaurants) of 38 municipalities.

### 2.2. Selection of Participants

The study involved key informants implicated in school feeding services in 21 municipalities in southern Brazil and the Popular Restaurant Program in 17 municipalities from all regions of the country (south, north, northeast, southeast and midwest).

For the selection of the 21 municipalities included, a probabilistic sample by conglomerates was performed among the municipalities of the southern region of Brazil. In order to cover key informants from municipalities of regions with different socioeconomic, geographical and cultural backgrounds, we randomly selected municipalities from different mesoregions. Mesoregions are subdivisions of states that group municipalities of a geographical area with economic and social similarities [25]. The inclusion criteria for this selection were a population between 20 and 70 thousand inhabitants, having a nutritionist as technical head of school feeding, and accepting participation in the research. The acceptance happened via telephone contact with the Education Departments of each selected municipality.

For the selection of the 17 municipalities included, we considered those that were listed and registered by competent institutions in the country. We used data from the Ministry of Social Development and Fight Against Hunger (MDS) website, which reported the presence of 100 popular restaurants throughout Brazil, distributed along 20 states. A convenience sample was carried out among the states, selecting the municipality with the largest number of inhabitants.

Participants were selected from a list of potential key informants previously defined by the researchers using a snowballing technique. The initial list was discussed among the researchers until a consensus was reached. The catering manager was initially contacted by email and/or telephone, and contact was facilitated with the other key informants.

### 2.3. Data Collection

Interviews were carried out between March 2015 and February 2017. Two hundred and eight interviews were carried out, grouped into three categories based on experiences and work area: A. Food production (*n* = 79), B. Management and marketing (*n* = 46), and 3. Meal production process (*n* = 83). Table 1 shows the reporters included in each group.

Nutritionist researchers trained during field visits in the workplace of the study participants conducted the data collection. Before the data collection, the participants received information about the study and were assured anonymity. All study participants signed the informed consent form (ICF). The research was approved by the Ethics Committee of the Federal University of Santa Catarina (UFSC) under protocols nº 1.002.956 and 13.314.367.

### 2.4. Instrument for Data Collection

The instruments for data collection were composed of semi-structured questionnaires regarding the food supply to public food services. These questions were prepared by the research team, nutrition academics, and nutritionists, with the collaboration of experts on the subject. They were tested in a previous pilot study, where one of the nutritionists applied the instruments to key informants of a public school and a popular restaurant. For the present study, an open question was selected for analysis: “In your opinion, what is quality food?”. The interviews were recorded, with the consent of the interviewees, and transcribed literally.

### 2.5. Data Analysis

We used Version 11 Pro of the Nvivo software for data analysis and organisation.

For data analysis, the Content Analysis method [26] is complemented by the coding in cycles [27]. The analysis was performed in two coding cycles. The first coding cycle was carried out by two researchers, coding themes and nuclei of the answers to the research question that offered meaning to the term “quality food”. After that, a discussion was held for consensus among the researchers, differentiating, organising and thereby highlighting the relevant codes. Then, the second coding cycle was performed, in which the data was reanalysed for a conclusion of the analysis.

For the categorisation of the codes, the semantic criterion (conceptual or thematic) was adopted as a classification criterion [26]. In addition, we opted for the mixed categorisation model, which groups the codes into categories formulated according to the initial hypotheses of the research (a priori), allowing for modifications in the course of the analysis [28]. Thus, the codes are grouped into generalist categories, formulated in the pre-analysis according to the research hypotheses and regrouped into terminal categories at the end of the analysis. We adopted the quality dimensions associated with food as generalist categories. Based on the literature, six categories of analysis were defined a priori: (1) hygienic–sanitary; (2) nutritional; (3) regulatory; (4) sensory; (5) symbolic; (6) sustainable [2,5,6,7].

To identify the most frequent category, we counted the number of text fragments encoded in each category. To explore possible differences in the opinion of the research participants about what quality food is, the results were stratified according to the professional experience of each of the participating groups (Groups A, B and C).

## 3. Results

The data analysis resulted in 52 codes grouped into eight categories, expressing the opinion of the study participants about what quality food is. Six of those referred to the existing categories at the beginning of the analysis and two emerging categories: a dimension of care and a dimension of dependability. Table 2 shows the terminal categories with the respective codes and lines that exemplify them.

The nutritional dimension grouped responses related to the guarantee of meeting the nutritional needs of individuals, highlighting the balance of food, variety and nutritional adequacy. The sustainable dimension was more related to forms of food production, emphasising quality as organic, local, fresh and seasonal foods. The sensory aspects of quality were divided into extrinsic attributes (such as appearance) and intrinsic attributes (such as maturation), and they were influenced by the way of preparation. This dimension included opinions that related food to the result triggered by human senses, arranged by sight, touch, smell and taste.

In the hygienic–sanitary dimension, the responses were related to food safety, especially in offering food without health risks. The care dimension grouped responses related to the relationship of care in all the processes that food is submitted to, referring mainly to the relationship of respect, diligence, commitment and zeal in food production. In the regulatory dimension, key informants associated quality with accordance with legislation and regulations in production and consumption. In the dependability dimension, the answers were in the sense of the relationship of trust between producer and consumer as an important aspect of quality. The symbolic dimension was related to the cultural context of food.

Figure 1 shows the frequency of codes grouped by categories identified in the study that explain the opinion of the different groups of Key Informants on what quality food. By analysing the number of coded citations (*n* = 611), it was observed that the nutritional dimension was the most frequently mentioned (*n* = 182), followed by the hygienic–sanitary (*n* = 109), sustainable (*n* = 123) and sensory (*n* = 130) dimensions. When analysing the number of encodings in each dimension stratified by the key-informants group, we observed that the nutritional dimension was more frequent for Groups B (Management and Marketing) (*n* = 54) and C (Meal Production Process) (*n* = 84), and sustainable for Group A (Food Production) (*n* = 70). The symbolic dimensions, dependability, regulation and care were less frequent among all groups.

## 4. Discussion

This study explored the concept of quality food in the opinion of key informants linked to different stages of the food system. The participants recognised several aspects of food quality, and the greatest consensus was regarding the nutritional, sustainable, sensory and hygienic–sanitary aspects. To a lesser extent, we observed references to aspects of regulatory quality, care, dependability and symbolic nature. The results suggest a difference in the opinion of key informants according to their professional experience. In the present study, the group linked to food production placed more emphasis on the sustainable dimension, followed by the nutritional one. The groups involved with the management and marketing and with the productive process of meals emphasised the nutritional and sensory dimensions. All groups identified extrinsic aspects of food as an important attribute of food quality, with emphasis on cooks, who gave more importance to appearance.

The nutritional dimension appeared prominently in the statements of informants from all groups. However, there was greater evidence in the responses of those linked to the productive process of meals, especially nutritionists. This result may be related to the prioritisation of nutritional aspects in the training of these professionals [29,30]. However, the increasing global concern about the sustainability of the food system highlights the importance of the sustainable dimension for food quality [15] and the need to incorporate this perspective in the training and performance of the nutritionist professional.

According to the authors [31], there is a need to improve the quality of food that should be taken into account; how and by whom our food is grown, and the implications for biodiversity, for local employment, fair trade and social justice [32]. Eating habits, local culture [33,34] and culinary skills are also considered key aspects or even facilitators for more sustainable eating patterns [35,36].

Key informants linked to the productive process of meals also spoke of quality food as one that has adequate sensory characteristics. These expectations may be related to the area of activity of this professional since the raw material is directly related to the object of work of this position, and the requirements of the final consumer fall onto them. Our result is similar to those of previous studies that highlighted sensory quality as decisive for food acquisition by consumers [37,38].

However, it should be considered that the appearance of the food might be related to the form of production used. A good example is agroecology and organics, which are more respectful of the environment and produce more nutritious foods [39,40]. They do not prioritise food production with a focus on sensory aspects, as occurs in conventional food production [41], which can generate rejection of products by professionals who work directly with the food, such as nutritionists or cooks since they are used to manipulating food in a certain format and pattern. This result highlights the importance of the intersectional approach to the acquisition of agroecological and organic food, connecting health, agriculture and the environment with technology sectors so that they work together [42].

Although there is divergence among the participants of the research on quality aspects, the nutritional, sustainable, hygienic–sanitary and sensory dimensions seemed to have a certain degree of consensus, even if they were mentioned with different frequencies among the informants of each group. Specifically, the hygienic–sanitary dimension was present in the responses of nutritionists and cooks, who emphasised the importance of safe and contaminant-free food, as observed in [43].

It is worth mentioning that even if the symbolic dimension is present in the definition of food and nutritional security [44] and is incorporated into international food-related recommendations [4,24], it was not often highlighted by the research participants, and even so, the definition does not deal with quality. There are studies defending the idea that managers of public food and nutrition facilities should carry out strategic planning to adapt the food supply of the institution to the regional eating habits and food culture of the population being served [45]. They must balance nutritional aspects, symbolic, as well as sustainable aspects of food (organic, local food, from family farming, for example). Although the regulatory and symbolic dimensions were little relevant among the research participants, the importance of these dimensions in the discussion of quality is fairly known [3].

Although some dimensions have been commonly mentioned among the groups of the research participants, some key informants seem to associate quality according to their insertion stage in the system, considering their activities pertinent to this stage, such as cooks and nutritionists. A study showed that previous experiences and the context in which the individual finds themselves influence their opinion and reference to food quality [20].

It is important to mention that the two emerging categories (care and dependability) identified in the present study also deserve to be discussed in the context of the food system, especially among the actors of the system, for the delivery of food that further guarantees consumer satisfaction. It is known that, from the consumer’s point of view, issues related to credibility, trust, and care regarding food are among the determinants of the consumer’s food choice [46,47,48,49,50]. This may be due to episodes of food outbreaks, which have affected consumer confidence regarding food safety primarily [46]. This confidence or care for the food can take place after an approximation of all the actors of the food system with the consumer. This approach can be through greater availability of food information, especially through food regulatory institutions [50], or by the very rapprochement between consumers and actors, especially if they manage to do it personally [49]. In 2006, some authors identified, in the opinion of consumers, that farmers are the actors who manage to convey greater confidence about food [50]. Therefore, it is a game changer for consumer satisfaction that all actors of the food system are engaged and have as a principle the zeal for food in all stages of the food system and that they are able to communicate it to the consumer by being closer or giving more information about this food, for example.

One of the limitations of this qualitative study is that the results are based on the opinions of key informants. As such, they cannot be generalised. However, the selection of participants with different professional backgrounds allowed a first approximation to the opinion of key informants linked to different stages of the food system on the quality of food. 

## 5. Conclusions

Through the present study, we can infer that both the nutritional and hygienic–sanitary dimensions are more consolidated to define quality food, different from the other dimensions, which were not incorporated in their entirety by the key informants participating in the research. The activities carried out in the food system seem to influence the opinion on food quality. The group linked to food production placed more emphasis on the sustainable dimension, followed by the nutritional one. The groups involved with the management and marketing and with the productive process of meals emphasised the nutritional and sensory dimensions. These differences may indicate a difficulty in the transition towards a more sustainable and healthy food system.

## Figures and Tables

**Figure 1 nutrients-16-00948-f001:**
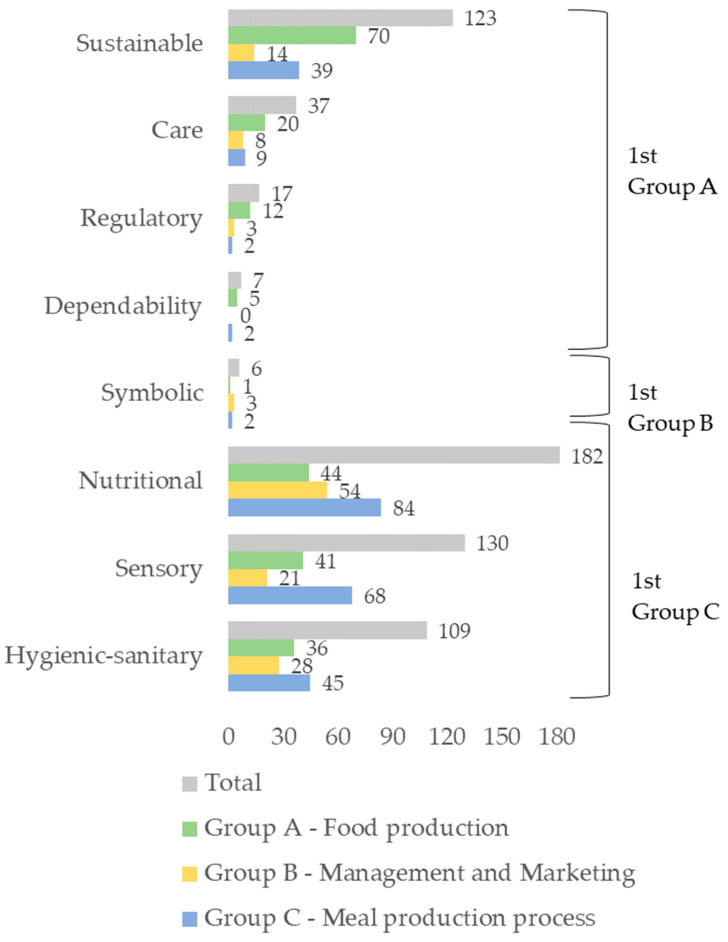
Frequency of codes grouped by categories identified in the study that explain the opinion of the different groups of Key Informants on what is a quality food (*n* = 611).

**Table 1 nutrients-16-00948-t001:** Participants of the study were divided into groups.

Group	Key Informants *n* = 208
Group A: Food production (*n* = 79)	Farmers (*n* = 43) Representatives of farmers’ cooperatives (*n* = 15) Agricultural technicians (*n* = 16) Agricultural Engineers (*n* = 5)
Group B: Management and marketing (*n* = 46)	Public Equipment Manager (*n* = 13) Secretaries of municipal departments linked to public equipment (*n* = 5) Representatives of the Food Council that public equipment is linked to (CAE) (*n* = 21) Directors of the public equipment the participant is part of (*n* = 4) Others (*n* = 3)
Group C: Meal production process (*n* = 83)	Cooks (*n* = 42) Nutritionists (*n* = 36) Others (*n* = 5)

**Table 2 nutrients-16-00948-t002:** Categories, codes, and examples of citations were provided according to the opinions of the survey participants.

Categories and Codes (*n* = 52)	Example
1 Nutritional (*n* = 9)	Meeting at least the nutritional values, the correct nutrients. (Nutritionist; Group C); Ah, a quality food is just … a healthy food, right? (Cook; Group C); A food that has everything we need, right? (Food Engineer; Group C); Ultra-processed food is not healthy food, is it? (Manager; Group B); I think it is the natural, right, a quality food is fresh, not so processed (Nutritionist; Group C); More variety, right? (Nutritionist; Group C) Real food, rice, beans, vegetables and fruit (Cook; Group C)
Nutritional composition according to recommendations	
Healthy food	
Meeting nutritional needs	
Minimally processed food	
Food in natura	
Offers variety in food items	
Real Food	
Offers food and nutritional security	
Offers balance between food groups	
2 Sustainability (*n* = 10)	A food without products, without poisons (Farmer; Group A); Preferably produced in the municipality or in the same region (Agronomist; Group A); An organically produced food, without the use of pesticides (Agricultural technician; Group A); It has to be produced in a way that does not harm the environment (Agricultural technician; Group A); It is the food that is natural, of the season (Nutritionist; Group C)
Pesticide-free	
From local production	
Organic	
Sustainably produced	
Seasonal	
Agroecological	
Produced in the field	
Sourced from diversified production	
Full use	
Fresh	
3 Sensory (*n* = 15)	It has to be well maintained and good-looking (Cook; Group C); A good-looking food, beautiful in color, bright, with good appearance (Cook; Group C); I think for me it is when it comes in a closed package, it comes correctly (Representative of the school feeding board; Group B); I think it comes well presented to the customer! We are concerned that it is well-presented, a very special thing (Public Equipment Coordinator; Group B); I think a quality food has to have all the proper sensory characteristics (Nutritionist; Group C); Firstly, it can have no defects; it is the first thing I notice! (Nutritionist; Group C)
Has proper appearance	
Colorful	
Intact packaging	
Has adequate presentation at the time of consumption	
Has adequate sensory characteristics	
In perfect condition	
Standard sized	
Has no relation to appearance	
No relation to appearance	
Tasty	
Factors related to maturation	
Characteristic texture	
Durability	
A characteristic smell	
Well elaborated	
4 Hygienic–sanitary (*n* = 9)	It is one that we can prepare without posing a risk to our health (Cook; Group C); Quality is everything from cleanliness itself, to quality and cleanliness is essential (Representative of the Cooperative; Group A); Not having any part, like, with worms in it, right? (Stockist; Group C); I think food from a safe source! The minimum quality and safety of the raw material (Nutritionist; Group C); A food item that has already left the farm with quality, so that it does not lose the quality. When storing this product, if it is to be frozen, refrigerated, there needs to be a cold room (Farmer; Group A)
Safe	
Hygienic	
Free of bacteria, pests and fungi	
Produced with safe raw material	
Respects food storage and conservation methods	
Transport compliance to ensure safe food properties	
It needs to be within the expiration date	
Compliance with sanitary standards of consumption	
Complies with sanitary standards of handling	
5 Care (*n* = 3)	Quality food starts from the ground, right? It is in the care, management, use of products that are not aggressive to the environment and to humans (Secretary; Group B); A safe food, with care in its handling and production. (Nutritionist; Group C)
Sourced from an agricultural production with careful planting	
Handled carefully	
Planted with care	
6 Regulatory (*n* = 2)	There are products (pesticides) that you cannot use because the shortage is very high (Farmer; Group A)
Produced according to the technical standards of agricultural production	
Manipulated according to standards	
7 Dependability (*n* = 2)	A quality food is one that we know its origin, right? How it was produced, how it was harvested (Agricultural technician; Group A)
The consumer has knowledge of food production and origin	
Direct contact with producer	
8 Symbolic (*n* = 2)	It is a food that respects the food culture (Nutritionist; Group C)
Respects food culture	
It has meaning	

## Data Availability

The raw data supporting the conclusions of this article will be made available by the authors on request.

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
