# Peer review of "What Is Quality Food? The Opinion of Key of the Brazilian Food System"

_nutrients, 2024, doi:10.3390/nu16070948_

Round 1

Reviewer 1 Report

Comments and Suggestions for Authors

The article is clear and concise, my only comment is on Table 2 and Figure 1. Figure 1 seems to be essentially an extract from Table 2 - therefore duplication of information - although visually interesting. I suggest choosing one method of processing - and putting all the information into it.

Author Response

Response: We appreciated your time in reviewing this work. In this new version, we have incorporated the suggestions of the reviewers, and we believe that this has contributed to improvements in the manuscript.

The article is clear and concise, my only comment is on Table 2 and Figure 1. Figure 1 seems to be essentially an extract from Table 2 - therefore duplication of information - although visually interesting. I suggest choosing one method of processing - and putting all the information into it.

Response: We agree with the comment. We have withdrawn figure 1.

Reviewer 2 Report

Comments and Suggestions for Authors

To explore the concept of quality food in the opinion of key informants of the food system, the participants were grouped into 3 groups according to their participation in the food system: 1. Food production; 2. Management and marketing; 3. Meals production process. Key informants answered the following question In your opinion, what is quality food?. These differences may indicate a difficulty in the transition towards a more sustainable and healthy food system. The study is meaningful. The experiment design is good.

Specific concerns:

1. In the part of “2. Materials and Methods”: It is necessary to divide these part into different subheading.

2. In the part of “2. Materials and Methods”: some description should be transferred to the part of 1. introduction.

3. Table 1: It is meaningless to have only one sample as Public Equipment Managers Coordinator (n=1), Social worker (n=1), Food and nutrition security coordinator of the municipality (n=1), Nutrition technician (n=1), Stockists (n = 3).

4. Please unify the format of references.

Comments on the Quality of English Language

Minor editing of English language required

Author Response

Response: We appreciated your time in reviewing this work. In this new version, we have incorporated the suggestions of the reviewers, and we believe that this has contributed to improvements in the manuscript.

Specific concerns:

  1. In the part of “2. Materials and Methods”: It is necessary to divide these part into different subheading.

Response: the Materials and Methods section has been divided into different subheading.

  1. In the part of “2. Materials and Methods”: some description should be transferred to the part of “1. introduction”.

Response: Description from the school feeding program and the popular restaurants has been transferred to the introduction. Please, see the fifth paragraph of the introduction.

  1. Table 1: It is meaningless to have only one sample as Public Equipment Managers Coordinator (n=1), Social worker (n=1), Food and nutrition security coordinator of the municipality (n=1), Nutrition technician (n=1), Stockists (n = 3).

Response: Respondents from categories with fewer representatives have been grouped into others category. Please, see table 1.

  1. Please unify the format of references.

Response: the format of references has been revised. Please, see the references section.